# Mental toughness in the Football Association Women's Super League: Relationships with playing experience, perceptions of club infrastructure, support mechanisms and self-esteem

Clare Wheatley[1], Mark Batey[2]*, Andrew Denovan[2], Neil Dagnall[3]

**1** Arsenal Women Football Club, London, United Kingdom, **2** Department of People & Performance, Manchester Metropolitan University Business School, Manchester, United Kingdom, **3** Department of Psychology, Manchester Metropolitan University, Manchester, United Kingdom

☯ These authors contributed equally to this work.

* m.batey@mmu.ac.uk

**Data Availability Statement:** Data cannot be shared publicly due to concerns of participant

## Abstract

Previous research reports a positive association between possession of mental toughness (MT) and high performance in sportspersons. However, the extent to which MT is related to playing experiences and appreciation of club environment in elite women's football has received only limited research attention. Accordingly, the present study investigated MT in the context of the English Football Association Women's Super League (WSL). Specifically, this paper examined relationships between level of MT and external (playing experience, perceptions of club infrastructure, and appreciation of support mechanisms) and internal (self-esteem) factors. A sample of 63 elite female professional football players from the WSL, aged between 18 and 35 years (mean = 25.87, $SD$ = 4.03), completed self-report measures. To objectively validate self-ratings, congruence between self and peer-rated was assessed. This revealed a strong degree of consistency. Subsequent analysis found positive correlations between MT, playing experience (number of years playing football, NoY; and highest level of football achieved, HLA), and External Support. Additionally, Self-Esteem correlated positively with MT, NoY, HLA, and External Support. Moderation analysis found MT interacted with NoY and predicted greater levels of Self-Esteem. Players with lower and mean MT, and more years as a professional were more likely to possess higher Self-Esteem (vs. less years). These outcomes indicated important relationships between MT, External Support, and Self-Esteem. Accordingly, WSL clubs can potentially apply the results of this study to enhance positive player mindset.

## Introduction

The Football Association Women's Super League (WSL) is a professional elite football competition in England, consisting of 12 teams. The WSL has experienced significant growth on and

anonymity. The athletes in this study are from an extremely rare and high profile population - female professional players in the Football Association's Women's Super League. The demographic data in the data file would lead to the athletes in the study being easily identified. This would be in breach of the ethical approval process and the briefings the athletes received about access to their data. Data are available from the Manchester Metropolitan University Research Ethics and Governance Committee (contact via FOBLEthicsEnquiries@mmu.ac.uk) for researchers who meet the criteria for access to confidential data.

**Funding:** The authors received no specific funding for this work.

**Competing interests:** The authors have declared that no competing interests exist.

off the pitch following the professionalisation of the league in 2018. Correspondingly, there have been increases in fan engagement, revenue, media interest, and sponsorship [1]. Expansion has resulted in heightened pressure and expectation being placed upon players, which potentially impacts their well-being, mental health, and wellness [2]. Investment in the WSL has led to the rapid development of club infrastructures and the allocation of budgets to support player development and achievement. One area clubs seek to develop because it positively correlates with high performance across a range of sports is player mental toughness (MT) [3]. However, there is scant research on the construct in elite women's football and no prior studies have investigated predictors of MT in elite women's football. Academic work in this area has been limited because the WSL is a highly-pressured environment, with a strong emphasis on competition and performance. Consequently, clubs have only limited resources (time and financial) to devote to scholarly investigation.

Noting the absence of MT work in the area of MT and that WSL players are an understudied, hard-to-access population, the present study explored MT in the context of the WSL. The specific purpose being to provide an understanding of the relationship between player MT and experience of key external and internal factors [4]. External factors were playing experience and perceptions of club infrastructure and support mechanisms. Internal factors were players perceived general self-esteem. The selection of factors was informed by the need to provide evidence on MT development to support the budgetary and resource allocation of the first author, whose role was to create a high-performance environment in a WSL club.

The construct of MT was conceived in the 1980s following work with elite athletes, which identified key characteristics allied to sporting success [5, 6]. Based on these attributes, Loehr [5, 7, 8] conceptualised MT as a psychological resource that facilitated stress tolerance and maximised an individual's capacity, regardless of circumstances, to perform consistently towards the upper range of their abilities. Accordingly, high levels of MT denote possession of efficacious psychological attributes such as self-belief, persistence, control, and effective mental skills [9, 10]. These are accompanied by complementary values, attitudes, emotions, and thoughts that assist goal achievement [11, 12]. Since these factors are not directly related to intellectual capabilities [13] they are commonly referred to as non-cognitive skills.

This classification acknowledges that attributes fall outside the domain of cognitive aptitude tests (i.e., intelligence scales) [14, 15]. Some theorists conceptualise non-cognitive skills as a mindset or allied beliefs that shape how individuals make sense of the world [16]. The term non-cognitive, however, is misrepresentative to the extent that these skills indirectly draw upon perceptive processes [14, 15]. From a performance perspective, MT manifests as the capacity to thrive in difficult situations and manage adversity (i.e., actively approach, respond to, and appraise demanding conditions). Thus, in a practical setting, MT is an adaptive psychological resource, which moderates the adverse consequences of pressure by mobilizing positive action and facilitating effective rebalancing following failure [17, 18]. Hence, high MT is linked with psychological benefits such as stress resistance and reduced depression [19, 20]. The notion that MT is a moderator of stressors is consistent with the construct's overlap with components of hardiness (i.e., challenge, commitment, and control) [21].

Hardiness is a personality disposition that has an established, empirical tradition as a protective buffer against harmful stress and concomitant ill-effects [22, 23]. It achieves this via the ability to transform demanding situations and circumstances into opportunities for growth and development [24]. Although, definitions of MT incorporate these features, especially the capacity to effectively manage adversity, overemphasis on these features is conceptually limiting because they are also present in other non-cognitive constructs such as resilience [cf. 24, 25]. The additional elements within MT are cognitions, emotions, and actions that help individuals achieve in positive circumstances [26].

Viewing MT as a general enabling psychological resource that promotes positive mental health, researchers have applied the construct to a range of achievement settings (i.e., sport [27], health [28], occupational [29], and educational [30]). Notwithstanding scholarly conjecture on the precise features of MT, the construct has consequently become an important non-cognitive skill within the domain of positive psychology and performance [12, 21, 31, 32]. The contribution of MT, however, has been undermined by conceptual disagreements, which have resulted in the adoption of differing definitions, models, and measurement instruments. Divergences centre on dimensionality (unidimensional vs. multidimensional), context-dependency (applicability to general vs. specific situations), and stability (trait vs. state) [33]. Although, these issues are beyond the remit of the current article it is important that researchers/practitioners possess awareness of debates (see reviews by Gucciardi [33] and Lin et al. [32]; and a commentary by Gucciardi [34]).

In the context of the present study, Clough et al. [21] stated that "Mentally tough individuals tend to be sociable and outgoing; as they are able to remain calm and relaxed, they are competitive in many situations and have lower anxiety levels than others. With a high sense of self-belief and an unshakeable faith that they can control their own destiny, these individuals can remain relatively unaffected by competition or adversity (p. 38)". This delineation is pertinent to the present paper since it recognises both the foundations of MT and the construct's applicability to real-world settings.

## MT and sport

Regarding sport, preceding research suggests that environmental factors are important to MT development, or at least to the formation of commensurate attitudes, cognitions, and dispositions. For instance, Bull et al. [35], following analysis of retrospective interviews with elite English cricketers, concluded that factors such as upbringing (i.e., parental influence and childhood background) and transition (junior playing career) were crucial to the growth of MT. This was also true of learning through experience (i.e., experiencing failure and encountering demanding environments).

Relatedly, Connaughton et al. [36], via retrospective evaluation of semi-structured interviews with elite athletes, explored the developmental template of MT (early = mean age of 8.3 years; middle, 11.1 years; and later years, 13.7 years). They observed that initial and early involvement (i.e., training once a week) provided a foundation for MT growth. Then, middle years (i.e., engagement in structured competition and training a few times a week) offered important formative experiences that further progressed MT. Particular instances being increased competition pressure and setbacks. These afforded opportunities to learn from mistakes and recover from setbacks. Later years (becoming fully committed to sport and training most days of the week) advanced MT by requiring the application of psychological skills and strategies. Consistent with these findings, Nicholls et al. [37] found that increased age and experience predicted higher levels of MT in a mixed sample of athletes. Moreover, Sheard et al. [38] reported significantly higher levels of MT in athletes aged 25 years or older (vs. younger athletes, 16–18 years).

The importance of environment was also reported by Gucciardi [39], who observed significant differences in MT between developmental groups of aspiring Australian Football players as a function of engagement with the sport. Investors (solely involved in Australian football) scored higher than specializers (those involved in Australian football during the Winter but also participating in a secondary sport during the Summer). Particularly, they reported higher levels of sport awareness (comprehension of team and individual performance) and desire success (achievement). Differences on sport awareness suggest that player engagement with feedback and self-reflection play an important role in MT development.

Golby and Sheard [40] assessed the extent to which personality style and mental skills predicted success in professional rugby league (i.e., players from the top three playing levels in Great Britain: International, Super League, and Division One). Players performing at the highest standard (International players) scored significantly higher on hardiness subscales (commitment, control and challenge) and two dimensions of MT (i.e., negative energy control and attention control). Negative energy control refers to the ability to manage undesirable emotions, and attention control denotes the ability to appropriately focus, maintain, and shift awareness.

Sheard [41] found that tournament winners (Australian Universities players) scored higher than their opponents from Great Britain on MT overall. Differences were also observed on specific MT factors Positive Cognition, Visualization, and the Hardiness dimension of Challenge (the degree to which players consider change to be normal and an opportunity for growth and development). Positive Cognition is the ability to thrive in adversity, specifically stay positive and to maintain a sense of fun and enjoyment when facing challenging situations. Visual imagery measures the athletes' use of positive visualization skills in training and competition. These findings concurred with previous research indicating superior MT and hardiness are related to successful sport performance.

Gucciardi et al. [26] using a personal construct psychology (PCP [41]) and grounded theory analysis identified three important categories related to MT with Australian Football League (AFL) players. These were characteristics (global attributes such as work ethic, self-belief, and personal values), situations (i.e., general and competition), and behaviours (also divided into general and competition). Viewed within this framework, MT acts as a buffer against adversity and helps to promote/maintain adaption to challenge.

Although, there have been myriad studies investigating the importance of MT in sports performers, relatively few studies have considered the role of the construct within footballers. One notable example is Crust et al. [42] who looked at MT in an English Premier League football academy (players aged between 12 and 18 years). They found no differences between age groups and those retained (vs. released). Additionally, Wieser and Thiel [43] administered a modified version of the Sports Mental Toughness Questionnaire and Psychological Performance Inventory to 20 male professional footballers and two coaches, who independently rated each player. Individual scores correlated positively, however, there was no agreement between player self-assessment and ratings by coaches. Footballers who played or had played for national teams (vs. not) achieved slightly higher mental hardiness scores.

Regarding triangulation of perceptions of MT, Gucciardi et al. [44], with a sample of Australian football players, observed no significant group differences between self, parent, and coach ratings. However, correlational analyses revealed only low to moderate relationships among raters. This is outcome was consistent with research in other contexts (e.g., organisational) where self vs. other rater agreements have demonstrated low to moderate, yet positive, relationships [45].

Preceding work on MT is also restricted by a limited focus on female athletes. This group in comparison to male performers are underrepresented within MT literature. Studies focusing on MT female sportspersons have produced important findings that have extended conceptual understanding of the role that MT plays in performance. For instance, Madrigal, et al. [46], using female athletes who participated in roller derby and collegiate rugby, found MT was both related to adaptive coping and positive injury response, and engaging in activity when injured. Another example is Wilson et al. [47], who through conducting semi-structured interviews with Canadian elite women athletes, identified that MT was critical for coping with sport-related adversity. Particularly, participants experienced MT as a coping resource, reflecting perseverance, maintaining perspective, satisfactory competition preparation, and focusing/

re-focus attention to the present moment. In addition to MT, common humanity, mindfulness, and self-kindness facilitated the capacity to cope with sport-related adversity. Furthermore, MT and self-compassion were perceived as compatible and contextually related processes. Self-compassion was vital to the development of mental toughness, and mindfulness was key to developing and maintaining self-compassion and MT. Findings from studies such as those conducted by Madrigal et al. [46] and Wilson et al. [47] are conceptually important because they broaden understanding of MT by providing insights into the way in which female elite athletes experience, perceive, and conceptualize MT. This is essential if sporting practitioners are to effectively use MT to enhance performance and well-being in sportspersons.

Despite the existence of some MT studies that have used female samples, there is a relative absence of related research in the area of women's football [48]. Kristjánsdóttir et al. [48] conducted a study of Icelandic women football players. Players were classified into three groups according to their level (i.e., national team, first, and second divisions). The national team had the highest score on MT and scored lowest on anxiety. This indicated that experience was a key factor in the development of MT. In another study, Williams et al. [49], observing that women football players in South African first division leagues possessed moderate levels of MT, proposed that players would generally benefit from psychological skills training as this would increase self-esteem and anxiety management.

## Present study

Noting the contribution of MT to sporting settings and the lack of studies investigating the construct's role in elite female football, the present paper examined relationships between level of self and peer-rated MT and factors associated with MT development in Women's Super League (WSL) footballers. Consistent with Aryanto and Larasti's [4] analysis of preceding research, this involved sampling external and internal factors. Since opportunities to access WSL were limited due to professional commitments the researchers focused only on key factors. These were identified via consideration of extant literature, and by drawing upon the lead author's experience as a general manager with a WSL club. Further constraints arose following the outbreak of COVID-19 and the ensuing restrictions.

External factors were divided into playing experience (years as a professional and highest level of international representation), perception of organisational infrastructure (i.e., head coach leadership style and learning culture), and support (within the club, internal, and outside, social). These factors sampled positive mindset, engagement, and ability to actively seek assistance both within and outside of the sporting organisation. The inclusion of playing experience derived from preceding research, which had found relationships between MT and playing history (i.e., international experience) [43, 48]. Perception of organisational infrastructure and support mechanisms indexed important factors related to MT growth and development. In the case of head coach leadership style and learning culture, positive mental environment (e.g., positive and confident atmosphere), and the provision of learning opportunities often compliment and facilitate attributes associated with MT [50]. Social support evaluated the extent to which players feel able to draw on their personal network to enhance functioning and/or buffer adverse outcomes [51]. The ability to seek reassurance and support outside of the club's infrastructure reflects active coping.

Regarding internal factors the researchers included global self-esteem. This decision was predicated on the observation that self-esteem is related to core features of MT (e.g., confidence and self-belief) [4]. Global self-esteem is the evaluative component of self-knowledge. Accordingly, high self-esteem indicates positive personal perceptions, whereas low self-esteem signifies less favourable evaluations [52]. Noting this, some theorists conceptualise self-esteem

generally as an important feature of 'positive thinking'. More specifically self-esteem influences attributional style, which denotes how individuals explain causation of positive and negative life events. This relationship is evidenced by studies, which report that self-esteem and positive attributional style are predictors of positive outcomes such as emotional well-being and academic performance. Relatedly, negative attributional style predicts a range of negative social and emotional outcomes [53, 54].

Hypotheses focused on MT, the degree of rater alignment, and the extent to which MT associated with evaluation of external and internal factors. It was predicted that there would be a significant positive relationship between self and peer rated MT (H1). Secondly, length of playing and representational level were expected to positively correlate with MT (H2). Moreover, the authors hypothesised (H3) that MT would be positively correlated with greater appreciation of infrastructure (i.e., head coach leadership style and learning cultures) and support (within the club, internal work, and outside, social). This was because individuals with higher MT should possess a positive mindset and perceive these factors as constructive mechanisms for self-development. Additionally, individuals with high MT are more likely to engage with available support because they are solution-focused and utilise active coping mechanisms [9]. Lastly, MT was expected to interact with length of playing and representational level and predict greater self-esteem (H4). Self-esteem is often treated as an outcome variable within sports research [e.g., 50], whereas MT, is considered to be more stable [55]. This relationship was predicated based on a previously established link between MT and playing experience [43, 48], alongside research demonstrating increased sports experience and MT are related with greater self-esteem [56, 57].

## Materials and methods

This study was cross-sectional and used correlation-based analysis to assess relationships between online, self-report measures.

### Participants

Participants were 63 female professional football players aged between 18 and 35 years (mean = 25.87, *SD* = 4.03). All played at elite level during the 2020/21 FA Women's Super League (WSL) season. In terms of highest level of international representation achieved: 4 (6.3%) = had not achieved international recognition, 1 (1.6%) = Under 17s, 10 (15.9%) = Under 19s, 11 (17.5%) = Under 23s, and 37 (58.7%) = Senior Honours. Regarding length of time as a professional, defined as the period during which a player had earned a full-time wage for training full-time: 10 (15.9%) = less than 12 months, 5 (7.9%) = between 12 and 23 months, 10 (15.9%) = 2 years to 4 years, 26 (41%) = 5 to 9 years, and 12 (19%) 10 years plus.

The lead author, to obtain as large a sample as possible, invited players from their club and the other WSL members (*N* = 162 players). Issues reduced the response rate, (i.e., changes in club management, *n* = 22; and disruptions arising from COVID, *n* = 38). Despite these factors, overall completion rate was 62%. Inclusion criteria were that players had to be at least 18 years old and be registered as a 'First Team' squad member.

### Materials

Self-report measures assessed level of mental toughness (MT), global transformational leadership, internal work support, external support, learning culture, and self-esteem. In addition, player MT was rated by two players from the same team. Each scale used a five-point Likert scale ranging from (1 = Strongly Disagree to 5 = Strongly Agree) unless specified otherwise.

**Mental Toughness Questionnaire 10-Item (MTQ-10).** Player level of MT was assessed using the MTQ-10 [55, 58]. Respondent provided a rating for self and two peers. The MTQ-10 is an abridged version of the MTQ-18 [21], which is a brief, unidimensional measure of MT. The MTQ-18 comprises high loading items from each of the four dimensions of the MTQ-48 (three Challenge, three Commitment, five Control, and seven Confidence). The MTQ-18 was developed because there was a need for a concise, global measure that researchers could expediently use in battery tests. Noting that the MTQ-18 was widely used, despite thorough psychometric evaluation, Dagnall et al. [55] examined the scale and found that although it was an adequate instrument, the MTQ-18 due to derivation from the MTQ48 was contaminated by additional variance arising from multidimensionality. Correcting for this produced the MTQ-10.

Within the MTQ measures, items are presented as statements (e.g., "I generally cope well with any problems that occur"). Peer ratings of MT were adapted, changing 'I' to 'she' (e.g., "Even when under considerable pressure *she* usually remains calm). Summation of items produces an overall score of MT. The MTQ-10 possesses promising psychometric properties. Specifically, good model fit via confirmatory factor analysis for a unidimensional solution, adequate composite reliability and test–retest reliability [58]. Additionally, the MTQ-10 has demonstrated concurrent validity and invariance (for gender: configural, metric, and scalar) [49].

**Head coach leadership.** The Global Transformational Leadership Scale (GTL) [59] evaluated the extent to which respondents believed their Head Coach positively facilitated constructive changes within players and social systems. The scale comprises 7-items indexing vision, staff development, support, empowerment, innovative thinking, leading by example, and charisma. Items within the GLT appear as statements and were adapted to change 'leader' to 'Head Coach' (e.g., "The Head Coach communicates a clear vision of the future"). Individual responses were totalled to produce an overall score, with higher scores indicating greater appreciation of transformational leadership. The GTL has established psychometric properties. Particularly, it has satisfactory reliability and validity [59].

**Internal work support (IWS).** Internal Work Support was measured using the Social Support section of the Work Design Questionnaire [60]. This comprises 6-items that index the degree to which a job provides opportunities for advice and assistance from others in the work environment. In the current study the focus was changed to club. Within the Work Design Questionnaire items are presented as assertions such as "I have the opportunity to meet with others in my work". Higher scores represent greater appreciation of internal support within the player's club. The IWS is a widely used measure with recognised psychometric properties (i.e., excellent reliability and convergent and discriminant validity) [60].

**Learning culture.** Learning Culture was measured using the Supportive Learning Environment (SLE) scale of the short-form Learning Organization Survey (LOS-27) [61]. This subscale assesses psychological safety, appreciation of differences, openness to new ideas, and time for reflection. Within the instrument, the 7-items appear as declarations. In the present study, items were modified from workplace to club (e.g., "In my Club, people value new ideas"). The focus of the items was whether players felt safe to express themselves and make errors, appreciated differences in opinions and ways of working, valued new ideas, and engaged in considered decision making. Higher ratings indicated a stronger evaluation of the club's learning culture. The LOS-27 and its subscales have established psychometric integrity (i.e., reliability and validity) [61].

**Social support.** The Brief Form of the Perceived Social Support Questionnaire (F-SozU K-6) [62] was used to evaluate player views of external social support. The scale conceptualizes social support as perceived or anticipated support from a social network. There are 6-items,

which are presented as statements. Within the present study these were adapted for use with players (e.g., "I experience a lot of understanding and security from others outside of work"). Higher scores signified greater appreciation of external support. The F-SozU K-6 has featured within several published studies and is regarded as a reliable, valid, and cost-effective psychometric instrument [62].

**Self-Esteem (SE).** The Lifespan Self-Esteem Scale (LSE) [63] measured players' subjective evaluations of the self. The LSE is a brief 4-item measure of global self-esteem suitable for populations drawn from across the life span. Within the instrument items appear as questions (e.g., "How do you feel about yourself?) with responses provided on a visual scale where 1 was an unhappy emoji face and 5 was a happy emoji face. The LSE is unidimensional, valid (e.g., possesses content and convergent validity), reliable (e.g., demonstrates internal and test-retest reliability), and converges with existing self-esteem measures across age [63]. Higher scale scores denote greater self-worth.

## Procedure

Following institutional ethical approval (Manchester Metropolitan University ETHoS Review #326130) and the distribution of the study details to each club contact by the lead author, respondents accessed information by clicking on a web link. Participants were asked to complete the survey part way through the season in November when 9 matches had been played. Only players who provided informed consent and met the inclusion criteria advanced to the self-report measures. Alongside rating scales participants completed a demographics section. This comprised: age (displayed as single ages), whether English was the player's first language or not, race/ethnicity, how many years they had played professional football for, primary playing position, the international age groups represented, whether the player featured in the most recent game for her club and had experienced any injuries this season. Peer ratings for the MTQ-10 were provided by team members of the player who played in a similar position, to ensure familiarity with the typical pressures of the role. Each player nominated two team members at the end of their survey, who were sent a separate survey link by email for their peer ratings. To limit common method variance, instructions encouraged psychological separation by stressing differences between constructs [64]. Respondents were debriefed after completing the measures.

## Results

### Interrater reliability

Comparison of self-rated MT ($M$ = 36.98, $SD$ = 6.10) and peer-rated MT scores ($M$ = 37.16, $SD$ = 6.33) within SPSS28 revealed no statistically significant difference, $t$ = .33 (55), $p$ = .744. In support of H1, calculation of the reliability coefficient between self-rated and peer-rated scores using intraclass correlation revealed a strong degree of consistency, $r$ = .88, $p$ < .001 [95% CI of .80 to .93]. The difference in scoring between self and peers appears in Table 1. A total of 30 exhibited identical MT ratings (54%), whereas a minority (3, 5%) exhibited a difference of 10 or greater.

### Preliminary analysis

Based on the similarity of rating types, subsequent analyses via SPSS28 used self-rating scores. Data screening of continuous study variables (MT, External Support, Head Coach Leadership, Internal Work Support, Learning Culture, and Self-Esteem) revealed no issues with skewness. All scores fell between ±3 [65]. Prior to testing, categorical variables of

**Table 1. Differences in scoring between self and peer MTQ10 ratings.**

| Scale (potential variation) | Number of agreements or disagreements in scoring |
|---|---|
| MTQ10 (0–50) | |
| Identical rating | 30 |
| ±1 | 4 |
| ±2 | 5 |
| ±3 | 3 |
| ±4 | 1 |
| ±5 | 2 |
| ±6 | 1 |
| ±7 | 5 |
| ±8 | 2 |
| ±10 or more | 3 |

number of years playing football as a professional (between less than 12 months and
10 + years) and highest level achieved (between under 15s and Senior) were transformed to
form categories (i.e., 6 years and less vs. 7+ years; under 23s and lower vs. Senior). The
median was used for this purpose. This strategy was consistent with previous related research
[44]. Subsequently, these group variables were dummy coded for the ensuing correlation and
moderation analysis.

Inspection of zero-order correlations using bootstrapping (1000 resamples) found, in support of H2, that MT was significantly, positively correlated with Number of Years Playing
Football (NoY) and Highest Level Achieved (HLA). However, though a significant association existed between MT and External Support, no meaningful correlations occurred
between MT and Head Coach Leadership, Internal Work Support, or Learning Culture. Collectively, these findings provided limited support for H3. Significant associations existed
between Self-Esteem and Number of Years Playing Football (NoY), Highest Level Achieved
(HLA), MT, and External Support (Table 2). Hence, these variables were used in moderation
analysis. Due to the small sample size, bootstrapping was implemented to provide less biased
estimates [66].

**Table 2. Descriptive statistics and correlations.**

| Variable | Mean | SD | Skew | 1 | 2 | 3 | 4 | 5 | 6 | 7 | 8 |
|---|---|---|---|---|---|---|---|---|---|---|---|
| | | | | | Correlation [95% bias-corrected and accelerated confidence interval] | | | | | | |
| 1. NoY | - | - | - | | .54** [.34,.75] | .33* [.09,.54] | .08 [-.17,.33] | .14 [-.12,.37] | -.06 [-.29,.19] | .05 [-.19,.26] | .41** [.22,.57] |
| 2. HLA | - | - | - | | | .25* [.03,.44] | .04 [-.18,.26] | -.03 [-.23,.19] | -.14 [-.35,.11] | -.17 [-.38,.06] | .41** [.22,.56] |
| 3. Mental Toughness | 37.11 | 6.24 | -.44 | | | | .25* [.04, .47] | .21 [-.04,.45] | .14 [-.09,.35] | .09 [-.16,.34] | .68** [.55,.79] |
| 4. External Support | 26.28 | 3.81 | -1.87 | | | | | .03 [-.25,.31] | .24 [-.02,.46] | .15 [-.13,.39] | .29* [.01,.53] |
| 5. Head Coach Leadership | 28.82 | 5.21 | -.84 | | | | | | .16 [-.09,.41] | .48** [.22,.70] | .11 [-.12,.37] |
| 6. Internal Work Support | 26.06 | 2.83 | -.45 | | | | | | | .38** [.16,.58] | .16 [-.05,.36] |
| 7. Learning Culture | 25.49 | 5.43 | -.39 | | | | | | | | .09 [-.16,.32] |
| 8. Self-Esteem | 16.76 | 2.29 | -1.16 | | | | | | | | |

*Note.* NoY = Number of Years Playing Football as a Professional, HLA = Highest Level Achieved;

* $p < .05$,

** $p < .001$ (using bootstrapping with 1000 resample

## Moderation analysis

As a test of H4, Hayes PROCESS macro for moderation analysis [67] assessed whether MT affected the strength and direction of the relationship between NoY and HLA with Self-esteem. This macro runs a series of OLS regressions with the centred product term representing the interaction of designated predictor and moderator variables with specific criterion variables. Analyses employed bootstrapping (1000 resamples) to generate 95% bias-corrected confidence intervals. Given its significant association with Self-Esteem, External Support was controlled for in the analyses.

Results indicated that NoY was a significant predictor of Self-Esteem, $b = 6.38$, 95% CI [.08, 8.05], $t = 2.04$, $p = .045$. MT was additionally a significant predictor, $b = .28$, 95% CI [.18, .37], $t = 6.01$, $p < .001$. A significant NoY and MT interaction existed, $b = .15$, 95% CI [.08, .28], $t = 2.11$, $p = .038$. External Support was not a significant covariate, $b = .08$, 95% CI [-.03, .18], $t = 1.44$, $p = .153$. Scrutiny of the interaction (via simple slopes) inferred that the relationship between NoY and Self-Esteem was significant at low ($t = 3.04$, $p = .003$) and mean ($t = 2.30$, $p = .024$) levels of MT (Fig 1). This indicated that a greater duration (in years) of playing football was associated with higher Self-Esteem, and this trend was particularly evident at low and mean levels of MT. The model accounted for 55.3% of variance (large effect size) in Self-Esteem.

HLA, however, was not a significant predictor of Self-Esteem, $b = 4.79$, 95% CI [-.02, 9.61], $t = 1.99$, $p = .051$. In addition, a significant interaction with MT did not exist, $b = .09$, 95% CI [-.22, .03], $t = 1.46$, $p = .147$. MT remained a significant predictor of Self-esteem, $b = .26$, 95% CI [.17, .35], $t = 5.83$, $p < .001$. These findings partially supported H4. Explicitly, outcomes indicated that MT interacted with length of playing but not representational level in relation to predicting Self-Esteem.

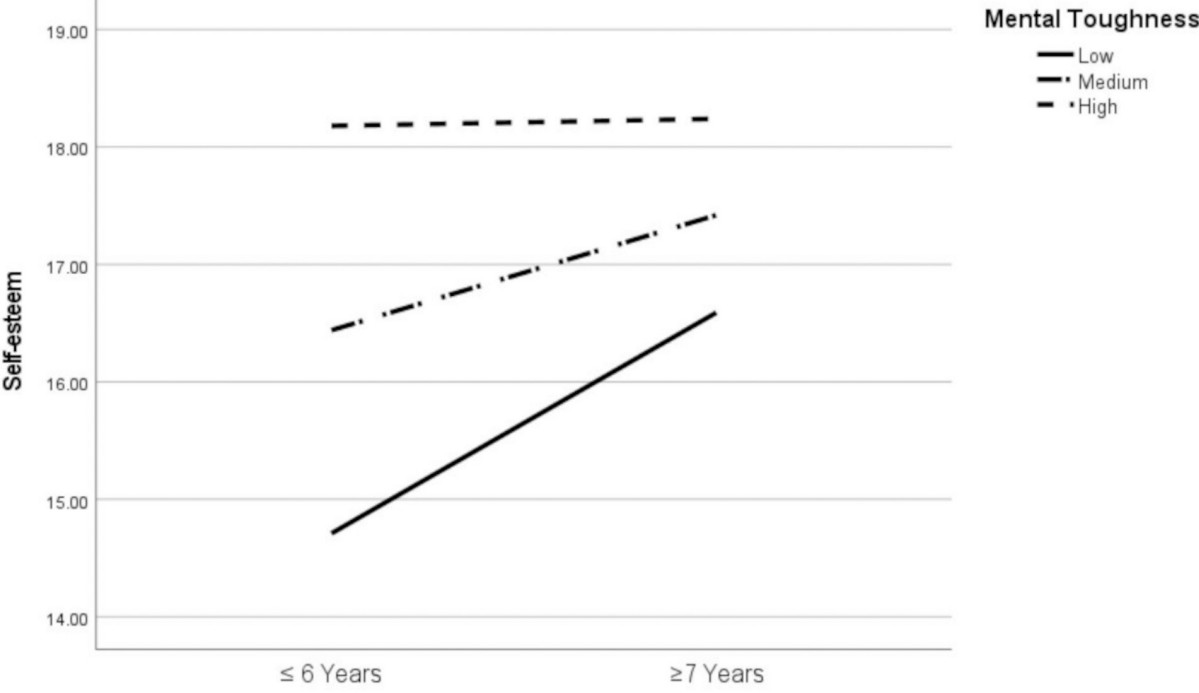

**Fig 1. Interaction via simple slopes of the relationship between number of years and self-esteem for different levels of Mental Toughness (MT).**

## Discussion

In support of H1, intraclass correlation revealed strong consistency between self and peer Mental Toughness (MT) ratings. It was necessary to establish the validity of self-ratings since critics of self-report methods have questioned whether individuals possess sufficient self-knowledge and insight to produce accurate judgments about their mental processes [68, 69]. Particular criticisms centre on lack of accuracy arising from subjective interpretations of complex psychological faculties [70], and the observation that high order functions are not fully available to consciousness [71].

A further issue is that self-ratings are prone to bias. Bias can occur unconsciously, as a consequence of subjective interpretation, or consciously in an attempt to produce socially desirable responses (faking good) or subvert test scores (faking bad) [72]. When error and bias occur, self-ratings are unlikely to be wholly representative of the construct or ability being assessed. At a measurement level this is problematic because it undermines validity. Acknowledging these concerns, the inclusion of peer ratings provided an index for objectively validating self-ratings. That is, an external assessment point, which was less susceptible to the potentially distorting influences of respondent subjective errors and bias (i.e., focusing on specific, atypical instances). In this context, the use of peer MT ratings authenticated self-reports by demonstrating that personal and external perceptions aligned.

The inclusion of peer ratings was also required because previous research, using football players from an English Premier League academy, reported differences between self (player) and other (coaches) assessment of MT [42]. This suggested that within elite level professional environments MT attributes and behaviours are often viewed differently. The agreement between self and peer ratings observed in the present study signified that FA Women's Super League (WSL) players shared similar observations of MT. From a psychometric perspective, consensus between rating scores and MTQ10 scores supported the construct validity of MT [69]. Notwithstanding this outcome, subsequent research should further explore the utility of MT peer ratings. This could include using multiple fellow players and coaches/managers. Triangulation between ratings would demonstrate commonality and stability across sporting viewpoints. It would also further assess whether the differences observed between players and coaches in preceding work extended to coaches/managers working within elite female football.

Supporting H2, examination of zero-order correlations revealed significant positive relationships between MT, playing-related factors (Number of Years Playing and Highest Level Achieved), and Self-Esteem. The magnitude of correlations was interpreted using the guidelines of Gignac and Szodorai [73] (i.e., small, $r = 0.11$; medium, $r = 0.19$, and large, $r = 0.29$). These derive from meta-analytical consideration of published research and represent typical effect sizes within real-world data. Accordingly, relationships between MT, playing-related factors and Self-Esteem were large. The pattern of results indicated that associations between psychological factors (MT and Self-Esteem) increased concomitant with playing-related experience.

This may be because psychological factors facilitated longer playing careers and higher levels of representation, or that professional football experience promoted the development of positive psychological factors. The finding that length and level of involvement in football was associated with higher MT and Self-Esteem aligned with previous sports-related studies that have identified experience as a central factor allied to MT (e.g., football [48] and cricket [35]). It concurred also with the supposition that relations were due to increased knowledge [3]. Explicitly, learning how to deal with difficulties, experiencing failure, and encountering demanding environments [74]. Support for the integral role of playing experience (vs. playing

achievement) was tentatively provided by the stronger relation between MT and Years Played (vs. Highest Level Achieved).

Noting the association between MT and experience, subsequent investigations should consider the incremental value that representative honours add to professional players. This could include assessment of the role of self-efficacy (i.e., an individual's belief in their abilities to achieve specific goals). Although self-efficacy is a discrete construct, within sportspersons it is related to both MT and accumulation of sporting knowledge [75, 76]. Additionally, the role of experience should be assessed using longitudinal approaches [77]. These studies would enable investigators to determine how MT develops over time. Longitudinal study would also facilitate identification of factors that moderate MT (e.g., injury, changes in team management, and perceived form) and provide an indication of their typical impact on players.

Within the present paper, the observation of a strong correlation between MT and Self-Esteem corresponded with prior investigations. Illustratively, Zeiger and Zeiger [57] within a sample of endurance athletes, reported large significant positive relationships between the three factors of the Sports Mental Toughness Questionnaire (SMTQ) (i.e., confidence, control, and constancy) and global self-esteem [38]. The association between MT and global self-esteem was attributable to construct overlap with self-belief [57].

Regarding self-esteem, it is important to recognise that the factor is multidimensional and that facets are differentially associated with outcome measures. Hence, in a sporting context, specific (sports-related) self-esteem is a better predictor of performance than global self-esteem (generalised self-worth), which is more strongly related to well-being [78]. Noting these differences, researchers use measures of global self-esteem to identify constructs that are indirectly (i.e., moderate/mediate) related to sporting performance [57]. In the current article, the importance of extending analysis beyond direct effects was demonstrated by moderation analysis. This revealed that Number of Years Playing Football as a Professional was associated with higher Self-Esteem, and this relationship was stronger at low and mean levels of MT. Players with lower and mean MT, and more years as a professional were more likely to have higher Self-Esteem vs. players with less years as a professional. This finding partially supported H4 and was consistent with research that reports associations between greater sports participation/experience and self-esteem [e.g., 56]. Additionally the outcome aligns with literature [e.g., 26] demonstrating that MT acts as a buffer against adversity (i.e., potentially protects self-esteem) in the sense that no notable Self-Esteem differences existed among females with high MT in relation to less vs. more playing years.

Global self-esteem was an appropriate measure to the extent that it placed an emphasis on general perceptions of organisation infrastructure and support. The expectation being that individuals higher in MT, due to a positive mindset, were more likely to constructively appraise organisation infrastructure and support since they provide opportunities for self-development, innovation, and challenge. However, both MT and Self-Esteem were only positively associated (medium-large effect) with External Support. The association between MT and External (Social) Support partially supported H3 and was consistent with preceding research [see 79]. Particularly, work indicating that social support moderates the effects of life stress, and that individuals with high (vs low) levels of social support are typically psychologically and physically healthier [80].

External Support overlap with MT and Self-Esteem may be explained by the fact that high levels of social support enhance an individual's active efforts to address sources of stress [79]. This supposition is consistent with Malecki and Demaray's [51] definition of social support as "an individual's perceptions of general support or specific supportive behaviours (available or acted on) from people in their social network, which enhances their functioning or may buffer them from adverse outcomes" (p. 232). Certainly, stress reduction and active coping are

ingredients of MT that are also likely to enhance self-esteem [81]. Although this notion is conceptually congruent, further work is required to develop models, and establish causal relationships.

There were no significant relationships between MT and Head Coach Leadership, Internal Support, and Learning Culture. A possible way to cultivate interactions between infrastructure and MT is to consider the contributory role of coaching style and climate, which previous research have identified as important factors [82]. Nicholls, Morley, & Perry [82] found that supportive coach behaviours were positively associated with task-involving climates, which in turn was positively associated with MT. The term supportive coaching refers to a broad, multifaceted coaching style that incorporates discrete yet interrelated emotional/relational and structural/instrumental components of effective instruction. The use of supportive coaching is important because the approach guides goal striving and nurtures development of athletic and mental skills.

In this context, supportive behaviours enhance athlete's self-efficacy, problem-solving, and ability to cope with stress [83]. The findings of Nicolas, Gaudreau, & Franche [83] indicated that this process is positively allied to MT through a task-involving climate. That is, a sporting environment where athletes perceive the focus of training to be skill mastery. The combination of supportive coach behaviours and task-involving climate create a setting in which effort and improvement are recognised and rewarded. Applied to the present study, this suggests that club infrastructure will increase levels of MT when coaches provide a reassuring, enabling environment that focuses on core skills and positively acknowledges endeavour and development. Reciprocally, positive perceptions of coaching style and purpose should heighten player appreciation of club infrastructure.

Another important factor that influences engagement with infrastructure is players' attributional styles. Meggs and Chen [84] found that high MT scores, within high performing swimmers, were positively associated with controllability. Specifically, possession of an attributional style, which viewed sporting failure as the result of internal, manageable factors. While self-perceived controllability is beneficial to the extent that it helps athletes maintain self-belief and perseverance, from a club perspective it may be problematic since individuals high in MT will generally be more independent and only seek support during times of transition and critical incidents [74]. This suggests that they are most likely to draw on club infrastructure when it is pertinent to their personal needs. Hence, sporting organisations, where possible, should attempt to create an accommodating infrastructure that aligns club facilities with individual demands [50]. One way to achieve this is to provide general services that are accessed via assessment of specific player needs. A concomitant advantage may be that players relate more strongly with the club because they see close alignment between personal and club identity. This is an important connection to nurture as the between club and player association can have a significant influence on performance and individual well-being [85].

The failure to find relationships between MT and infrastructure (Head Coach Leadership, and Learning Culture) and Internal Work Support, was contrary to the notion that players with high MT were more likely to constructively appraise these factors because they offer opportunities for self-development, innovation, and challenge. However, the findings of the present study should be treated cautiously due to the fact that provision varied greatly across clubs. Moreover, differences between club infrastructures were potentially exacerbated by unanticipated constraints (i.e., changes in club management and disruptions arising from COVID-19). Hence, future investigations should determine the extent to which player evaluations of head coach leadership, learning culture, and internal support are affected by variables such as the degree to which structures are embedded within the organisation, changes in team

management, length of time players have been at a club, duration of playing career, and level of seniority. Establishing this will enable an informed assessment of the potential role that MT plays in player engagement with organisational facilities.

Indeed, it is important to understand interrelationships between MT, external (playing experience, perceptions of club infrastructure, and appreciation of support mechanisms) and internal (self-esteem) factors because these variables collectively indicate which types of provision within sporting organisations are likely to maximise the psychological well-being and contentment of players. Additionally, the potential importance of infrastructure was demonstrated by large positive correlations between Learning Culture and Head Coach Leadership and Internal Work Support. These suggest that constructive appraisal of core club infrastructure elements are interrelated. Acknowledging the interrelatedness of internal organisational elements, preceding research should examine interactions further and examine how MT influences associations.

## Implications

The outcomes of this study have important policy implications for WSL clubs. Specifically, through coaching and player support services, clubs should place an emphasis on developing self-esteem. This is especially important within players with lower levels of confidence and less professional experience. Work in this area in academies would also help prepare young players for the demands of the WSL. In this context, MT is an important construct because it encourages a positive mental mindset, buffers against pressure, and facilitates the use of active solution-oriented coping mechanisms. The ability to seek control of external events and address issues/difficulties are important psychological attributes that promote and sustain higher levels of self-esteem. Equally, lower levels of self-esteem likely predict lower MT, passive, problem-focused coping mechanisms, and reduced levels of performance.

WSL clubs should also employ strategies that encourage the development and enhancement of external support networks. These are especially important during transitional periods, such as transfers between clubs and international representation, when players are less able to draw upon internal support mechanisms. Additionally, providing opportunities for players to bond and socialise offers assistance for players that extends beyond formal club provision. These activities are not only likely to facilitate MT and self-esteem but also promote trust and belonging in the team/club. Perceived alignment of self, others, and club values concomitantly should increase unity and common sense of purpose. Player 'buy-in' is essential if they are to utilise and act upon the outcomes of self-report measures. Explicitly, if they are going to view instruments as development tools that produce accurate and meaningful insights to improving and maintaining high levels of personal performance.

## Limitations

Although the present study was largely exploratory in nature, it makes an important contribution to research by investigating the role of MT in elite level football players in the WSL. Despite the rapid expansion of the Women's game in recent years this remains an under researched population. In this context, understanding the factors that influence player perceptions of club and personal support mechanisms is crucial to individual well-being and organisational efficacy. Notwithstanding this impact, findings require cautious interpretation due to study limitations.

One obvious concern is sample size, which was lower than expected due to the outbreak of the COVID-19 pandemic and ensuing restrictions. This greatly impacted upon recruitment since the principal investigators were not able to meet with teams to cultivate engagement and

concomitantly several clubs experienced re-organisations meaning that commitment to involvement faltered. One potential issue with small samples is that analysis is potentially underpowered. A second issue is that the margins of error increase and responses may not be generally representative. These factors can undermine validity and limit the generalisability of outcomes. However, it is worth noting that a large effect size was observed in the moderation analysis pertaining to Number of Years as a Professional.

In the present study although the sample size was relatively small, it still included a high percentage of the female players across the WSL; there are currently only 12 teams in the WSL with squad sizes of 23. Furthermore, due to organisational and professional constraints these are a difficult population to access. In this context, it is worth noting that MT-focused studies investigating specialist populations published within peer-reviewed literature have also used small samples. For example, Crust et al. [42] English Premier League academy football players (N = 112); McAuley et al. [86] football players from an English academy (N = 73); and Golby and Sheard [40] international rugby league players (N = 70). Similarly, studies focusing on related constructs (i.e., hardiness) have used relatively small, convenience samples [see 43]. Thus, while sample size is a concern there is no reason to believe that responses were not typical of WSL players. Despite this, given the exploratory nature of this study there remains a need to replicate and extend findings.

A further potential criticism is the use of the unidimensional MTQ-10 in preference to either more established or multidimensional measures. The MTQ-10 was selected for two reasons. Firstly, as clubs accommodated testing within their busy schedules and the construct was measured alongside several other variables a brief scale assessing MT was required. Hence, to facilitate club participation and player engagement brevity was necessary. Secondly, the MTQ-10 was selected in preference to the established MTQ-18 because it is psychometrically superior. Notwithstanding debates on dimensionality the MTQ measures have generally performed well within research [87].

Illustratively, following a systematic review of MT measures Farnsworth et al. [87] noted that the MTQ-48 was the most prevalently used scale and concluded that the MTQ-18 was one of the best measures. It performed sufficiently across indices (structural validity, internal consistency, and hypothesis testing). In comparison to the MTQ scales, the Sports Mental Toughness Questionnaire (SMTQ) [38] was used rarely and was insufficient with regards to overall internal consistency rating. Acknowledging that assessment of MT factors may provide further insights into players' attitudes, subsequent investigation could, constraints allowing, use the MTQ-48 [88].

Another potential limitation is that outcomes may be contextually restricted. This is a major concern since data collection occurred during the height of the COVID restrictions. Consequently, there were great disruptions to club infrastructure. This in part may explain the relationship between MT and External Support, and account for the lack of association between MT and internal measures. This is something that ensuing investigations should establish. Certainly, COVID measures provided a unique set of circumstances that may have influenced player responses. That stated, contextual effects are a common concern within psychological research and care is always required to ensure that outcomes are representative and generalised appropriately.

Finally, although the present findings were consistent with previous research on MT, self-esteem, and social support, the cross-sectional design, where data are collected at one point in time, precludes the inference of causal relationships. Acknowledging this, ensuing investigation should assess relationships across multiple time points using a longitudinal design. This would establish the stability of the relationships and provide causal insights. Despite this, the present paper suggests conceptual possibilities for model development.

## Acknowledgments

The authors wish to thank the athletes from The Football Association Women's Super League (WSL) who kindly provided their time to complete the research measures.

## Author Contributions

**Conceptualization:** Clare Wheatley, Mark Batey.

**Data curation:** Clare Wheatley.

**Formal analysis:** Andrew Denovan.

**Investigation:** Mark Batey.

**Methodology:** Mark Batey.

**Project administration:** Clare Wheatley, Mark Batey.

**Resources:** Clare Wheatley.

**Supervision:** Mark Batey.

**Visualization:** Andrew Denovan.

**Writing – original draft:** Clare Wheatley, Mark Batey, Andrew Denovan, Neil Dagnall.

**Writing – review & editing:** Mark Batey, Andrew Denovan, Neil Dagnall.

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
