## [Decision Letter · Decision Letter 0]

28 Feb 2023

PONE-D-23-01152Mental toughness in the Football Association Women’s Super League: Relationships with playing experience, perceptions of club infrastructure, support mechanisms and self-esteemPLOS ONE

Dear Dr. Batey,

Thank you for submitting your manuscript to PLOS ONE. After careful consideration, we feel that it has merit but does not fully meet PLOS ONE’s publication criteria as it currently stands. Therefore, we invite you to submit a revised version of the manuscript that addresses the points raised during the review process.

We look forward to receiving your revised manuscript.

Kind regards,

Ender Senel, PhD

Academic Editor

PLOS ONE

Journal Requirements:

Reviewers' comments:

Reviewer's Responses to Questions

**Comments to the Author**

1. Is the manuscript technically sound, and do the data support the conclusions?

Reviewer #1: Yes

Reviewer #2: Yes

2. Has the statistical analysis been performed appropriately and rigorously? 

Reviewer #1: Yes

Reviewer #2: Yes

3. Have the authors made all data underlying the findings in their manuscript fully available?

Reviewer #1: Yes

Reviewer #2: No

4. Is the manuscript presented in an intelligible fashion and written in standard English?

Reviewer #1: Yes

Reviewer #2: Yes

5. Review Comments to the Author

Reviewer #1: First, I would like to commend the authors for choosing to target women’s football as their group of interest – this is a population that is sorely underrepresented in mental toughness (MT) research. Second, I have selected "major revisions" because I believe the discusion section needs work, all other comments, I believe, are "minor" in nature. I believe this manuscript has potential to be a valuable addition to the MT literature and would be happy to re-review the manuscript should the authors choose to re-submit.

Minor points:

Throughout the manuscript, there are a number of times where sentences are started with “That”, “These”, “It” and words of the like without clarification on what is being referred to.

There is inconsistent use of the phrase “mental toughness” and the abbreviation “MT” throughout manuscript.

I suggest a thorough proofread prior to re-submission.

Abstract

Line 34-35. I am confused by the sentence “However, there exists only limited research on the contribution of MT to elite women’s football.” It is not clear what this means. Are you talking about MT contributing to performance? Are you talking about how MT interacts with other factors? Are you talking about MT’s relevance to the women’s form of the game?

Introduction

P3, L74. Different parentheses used.

P3, L75. “Based on these,…” What are you referring to here?

P4, L84-86. “Some theorists…” this sentence would be stronger with a reference added.

P4, L93-95. You should change the reference here to Clough et al. (2002), as Gucciardi (2017) discusses a different conceptualisation.

Conceptual misalignment is an issue here. You mention Gucciardi’s (2017) definition, however, use a scale that is a derivative of Clough et al.’s (2002) 4/6 C’s conceptualisation. I would recommend either removing the paragraph on P5, L116-120 or changing this to reflect the 4/6 C’s conceptualisation.

I feel as though a stronger point could be made regarding the lack female representation in MT research. You could also highlight some research that has focussed on female athletic populations. Some examples are:

Madrigal, L., Wurst, K., & Gill, D. L. (2016). The role of mental toughness in coping and injury response in female roller derby and rugby athletes. Journal of Clinical Sport Psychology, 10(2), 137-154.

Wilson, D., Bennett, E. V., Mosewich, A. D., Faulkner, G. E., & Crocker, P. R. (2019). “The zipper effect”: Exploring the interrelationship of mental toughness and self-compassion among Canadian elite women athletes. Psychology of Sport and Exercise, 40, 61-70.

P10, L229. It seems odd that H1 states there will be a strong positive relationship between self and peer-rated MT, when, in the introduction, you mention Wieser and Thiel’s study finding no agreement between player and coach ratings. If this is because you are focussed on peer-ratings? If so, you may need to mention this.

Methods

I really like the employment of a peer-assessor of MT. I believe it is a simple step that should be adopted as widely as possible in the MT literature in cross-sectional designs. Well done, authors.

P14, L345-347. Good idea to have similar positions assess MT.

I was unable find details on what program (e.g., SPSS, R) was used to run the analysis.

Results

It may be useful to clarify which MT scores you included in the moderation analysis (i.e., the self-report or peer-reported). Only because you mention how peer-assessment may provide more accurate assessment of attitudes and behaviours in the discussion.

Discussion

The discussion needs work. As it stands it is too vague and needs to incoporate more research and theory - this will add more punch to your conclusions. For instance, the paragraph beginning on P22, L523. It appears the main message is that MT may influence how one views their club infrastructure – which is an important finding and adds to the collective understanding of MT. However, vague suggestions for future research are offered instead of discussing how this finding exists within our current understanding of MT. You could have referred to Nicholls et al. (2016) who linked MT to perceptions of coaching, or discussed Meggs and Chen (2019) who explored MT and attribution style (these are just examples). I think adopting this approach will strengthen the discussion and highlight your findings more clearly. I recommend trying to integrate more MT research into each section of your discussion where you mention a key finding.

Refs:

Meggs, J., & Chen, M. A. (2018). Mental toughness and attributions of failure in high performing male and female swimmers. Journal of Human Sport and Exercise, 13(2), 276-284.

Nicholls, A. R., Morley, D., & Perry, J. L. (2016). Mentally tough athletes are more aware of unsupportive coaching behaviours: Perceptions of coach behaviour, motivational climate, and mental toughness in sport. International Journal of Sports Science & Coaching, 11(2), 172-181.

P19, 439-440. Re-word sentence. Perhaps replace “specifies” with “demonstrates.”

P19, L443-444. It is not clear what this sentence means.

P19, 460-461. It is not clear what this sentence means.

P22, L513-522. May be useful to mention Gucciardi et al. (2017). Particularly as it was study with an all female cohort.

P22, L523. It’s not clear why you have started this sentence with “intuitively.” Also, how might these attributes affect assessment of club infrastructure?

P24, L572-573. Does this need to be said? Weren’t all players who took part female?

Paragraph P25 L600. How might have COVID-19 impacted these findings?

Reviewer #2: Thank you for the opportunity to review this manuscript. The authors demonstrated the concept and related factors of "Mental Toughness" in WSL athletes. The study is well conducted and the employed methods are appropriate. Regarding the description of the manuscript in the paper, there were no excesses or deficiencies in the explanations of each variable in the background. I was concerned that the results included a description of the analysis methods, but if the journal allows it, I don’t feel that it is a problem. Also, since the paper consists of many research results, it would be easier to read if the current results were summarized in the first paragraph of the Discussion or the Conclusion. It will be of interest to readers of this journal, particularly researchers and practitioners involved in the sports community.

6. PLOS authors have the option to publish the peer review history of their article (what does this mean?). If published, this will include your full peer review and any attached files.

Reviewer #1: No

Reviewer #2: No

---

## [Author Response · Author response to Decision Letter 0]

6 Apr 2023

Please note - the response to reviewers has been attached as a file

---

## [Decision Letter · Decision Letter 1]

27 Apr 2023

Mental toughness in the Football Association Women’s Super League: Relationships with playing experience, perceptions of club infrastructure, support mechanisms and self-esteem

PONE-D-23-01152R1

Dear Dr. Batey,

We’re pleased to inform you that your manuscript has been judged scientifically suitable for publication and will be formally accepted for publication once it meets all outstanding technical requirements.

Kind regards,

Ender Senel, PhD

Academic Editor

PLOS ONE

Additional Editor Comments (optional):

Reviewers' comments:

Reviewer's Responses to Questions

**Comments to the Author**

1. If the authors have adequately addressed your comments raised in a previous round of review and you feel that this manuscript is now acceptable for publication, you may indicate that here to bypass the “Comments to the Author” section, enter your conflict of interest statement in the “Confidential to Editor” section, and submit your "Accept" recommendation.

Reviewer #1: All comments have been addressed

2. Is the manuscript technically sound, and do the data support the conclusions?

Reviewer #1: Yes

3. Has the statistical analysis been performed appropriately and rigorously? 

Reviewer #1: Yes

4. Have the authors made all data underlying the findings in their manuscript fully available?

Reviewer #1: Yes

5. Is the manuscript presented in an intelligible fashion and written in standard English?

Reviewer #1: Yes

6. Review Comments to the Author

Reviewer #1: I am pleased to say the authors have addressed the previous concerns I raised. They have clearly put in a lot of work to reshape the manuscript and should be commended for their effort.

7. PLOS authors have the option to publish the peer review history of their article (what does this mean?). If published, this will include your full peer review and any attached files.

Reviewer #1: No

---

## [Editor Report · Acceptance letter]

4 May 2023

PONE-D-23-01152R1 

Mental toughness in the Football Association Women’s Super League: Relationships with playing experience, perceptions of club infrastructure, support mechanisms and self-esteem 

Dear Dr. Batey:

I'm pleased to inform you that your manuscript has been deemed suitable for publication in PLOS ONE. Congratulations! Your manuscript is now with our production department. 

Kind regards, 

on behalf of

Dr. Ender Senel 

Academic Editor

PLOS ONE